# SALF & TALF: Optimized Loss Function and Drafting for Tree-based Speculative Decoding

## Abstract

Speculative decoding (SpD) has emerged as a promising approach to accelerate the slow autoregressive inference of large language models (LLMs). SpD leverages a lightweight draft model to propose candidate tokens, which are then verified in parallel by the target LLM. Recent advances in tree-based SpD significantly improve efficiency by drafting token trees, enabling the verification of multiple sequences at once. Given its strong empirical performance reported across numerous studies, tree-based SpD is rapidly becoming dominant. However, existing draft model training methods overlook the tree structure when defining the training objectives, causing their training and inference distributions to become misaligned. We address this limitation with a tree-aware loss function (TALF) that explicitly incorporates the tree structure into draft model training. Using trees generated by the target LLM, TALF aligns the draft model's predictions with the target across all branches, mitigating the misalignment. Further, we improve the tree construction process in drafting with stopping at low further gains (SALF). As drafting iterations search for potential high-probability tokens to add to the tree, we estimate aggregate probability gains. This estimate guides the stopping criterion for drafting, enabling us to balance computational cost against draft quality for maximum performance. Together, SALF & TALF deliver 15.6–39.4% and 6.5–24.4% end-to-end speedups over state-of-the-art SpD methods, EAGLE-2 and HASS, without altering the draft model architecture.

## 1 Introduction

As large language models (LLMs) become integral to numerous real-world services, it is of great social, economic, and environmental importance to perform LLM inference both rapidly and efficiently. LLMs generate tokens in an autoregressive manner, producing one token per decoding iteration, which is then used to generate the next. Each iteration requires loading hundreds of billions of parameters (Brown et al., 2020) from memory. This is especially problematic for edge devices or similar environments with limited batching; without batching to amortize the memory load costs, hardware utilization is severely damaged due to the memory bandwidth bottleneck (Park et al., 2024).

As an effective solution to this problem, speculative decoding (SpD) (Leviathan et al., 2023; Chen et al., 2023) has been proposed. At a high level, SpD employs a lightweight *draft* model to speculate the output of the *target* LLM. First, multiple autoregressive iterations are performed with the draft model to create a short sequence of tokens (i.e., draft). Then, the draft is verified by the target LLM, which decides whether to accept each token in the draft, in a single forward pass. As multiple tokens can be generated per draft-verify, SpD enables generation of the same output sequence with fewer target LLM iterations. Using orders of magnitude smaller draft models, substantial reductions in the end-to-end LLM inference latency can be achieved.

Among various SpD approaches (see §5), EAGLE (Li et al., 2024b) and its successors (Li et al., 2024a; Zhang et al., 2024b) show notable speedups and are widely integrated into mainstream LLM serving frameworks, such as vLLM (Kwon et al., 2023), TensorRT-LLM (NVIDIA, 2025), and SGLang (Zheng et al., 2024). Two key techniques constitute EAGLE. The former is its draft model architecture with a single Transformer decoder (Vaswani et al., 2017) layer. Recent work, including HASS (Zhang et al., 2025) and Griffin (Hu et al., 2025), further refines the training objectives to better train the EAGLE draft model (detailed in §2.2).

The latter is the adoption of *tree-based SpD* (Miao et al., 2024). Instead of producing a single linear sequence of candidate tokens, the draft model generates a tree that captures multiple branching hypotheses. This draft tree can still be verified in a single forward pass by replacing the standard attention (Vaswani et al., 2017) in the target LLM with tree attention (Miao et al., 2024). As a single tree encompasses multiple probable sequences in a compact form, it significantly increases the number of tokens generated per draft-verify ($\tau$). Following work, EAGLE-2 (Li et al., 2024a), further introduces a dynamic tree construction algorithm that leverages token probabilities from the draft model, modifying the tree structure to incorporate more high-probability tokens in the tree.

While tree-based SpD is becoming a standard technique as evidenced by numerous related studies (Zhang et al., 2025; Hu et al., 2025; Li et al., 2024b;a; Miao et al., 2024; Chen et al., 2024a; Spector & Re, 2023; Sun et al., 2023; Ankner et al., 2024; Cai et al., 2024; Svirschevski et al., 2024), previous approaches fall short of exploiting its full potential in both training and inference. In particular, prior draft model training methods only focus on the most probable tokens, whereas tree-based SpD demands exploring alternative token candidates.

To address this, we introduce a *tree-aware loss function (TALF)* for SpD. An ideal draft model would generate the same tree as the target LLM. Therefore, we make the target LLM construct a tree dynamically during training. At each tree node (token), we compute a cross-entropy loss between the next-token probability distributions of the draft and target models. We then aggregate this loss over the entire tree, guiding the draft model to generate target-aligned trees.

We also propose a novel dynamic draft tree construction algorithm with a conditional stopping criterion, named *stopping at low further gains (SALF)*. We develop methods to predict further gains from continuing drafting and to stop when the gains fall below a configurable SALF threshold. SALF offers speedups over existing dynamic tree construction methods by delivering a balanced solution that trades off tree optimality (Svirschevski et al., 2024) with drafting overhead. SALF and TALF together result in 2.16–3.48× end-to-end speedups for Llama-based models (Touvron et al., 2023; Llama Team, AI @ Meta, 2024; Deepseek-AI, 2025) in various tasks, improving upon EAGLE-2 and HASS by 15.6–39.4% and 6.5–24.4%, respectively.

## 2 BACKGROUND

$\mathbf{T}$ denotes the target LLM and $\mathbf{D}$ denotes a small draft model imitating $\mathbf{T}$. Values derived from $\mathbf{D}$ are marked with a superscript $\cdot^{(d)}$. At a time step $s$, we try to generate a token $x_{s+1}$ from the previous tokens $x_{1:s}$. For notational convenience, we ignore the initial prefix prepared for each text generation.

### 2.1 SPECULATIVE DECODING (SPD)

First SpD constructions by Chen et al. (2023) and Leviathan et al. (2023) used a lightweight draft model (e.g., a 4B LLM) to generate a short sequence of candidate tokens $(x_{s+1}^{(d)}, x_{s+2}^{(d)}, \cdots)$ following $x_{1:s}$ through multiple autoregressive drafting iterations. The candidates are verified with the target LLM (e.g., a 70B LLM) through parallel processing of $p_{s+1} \leftarrow \mathbf{T}(x_{1:s}), p_{s+2} \leftarrow \mathbf{T}([x_{1:s}, x_{s+1}^{(d)}]), \cdots$, computed together in a single forward pass. Based on the probability distributions $(p_{s+1}, p_{s+2}, \cdots)$, we decide whether to accept each candidate token. For example, when $p_{s+1}(x_{s+1}^{(d)})$ is likely but $p_{s+2}(x_{s+2}^{(d)})$ is not, $x_{s+1} \leftarrow x_{s+1}^{(d)}$, $x_{s+2}$ is sampled from $p_{s+2}$, and the verification ends. Two key metrics are used to evaluate SpD: 1) end-to-end latency of LLM inference, including times spent with $\mathbf{T}$ and $\mathbf{D}$, and 2) mean generation length ($\tau$), the average number of tokens generated per verification.

### 2.2 DRAFT MODEL TRAINING: EAGLE & HASS

EAGLE (Li et al., 2024b) designs a small draft model with a single Transformer decoder (Vaswani et al., 2017) block to accelerate SpD. During inference, *features* produced from the last decoder block of the target LLM ($f_{1:s-1}$) as well as $x_{2:s}$ (rather than $x_{1:s}$ so that $|x_{2:s}| = |f_{1:s-1}|$) are fed into the draft model. The draft model performs autoregressive iterations with these inputs; i.e., $p_{s+1}^{(d)}, f_s^{(d)} \leftarrow \mathbf{D}(x_{2:s}, f_{1:s-1})$, then $p_{s+2}^{(d)}, f_{s+1}^{(d)} \leftarrow \mathbf{D}([x_{2:s}, x_{s+1}^{(d)}], [f_{1:s-1}, f_s^{(d)}])$ with $x_{s+1}^{(d)}$ sampled from the speculated probability distribution $p_{s+1}^{(d)}$, and so on. Verification of the candidate tokens are handled in the same way.

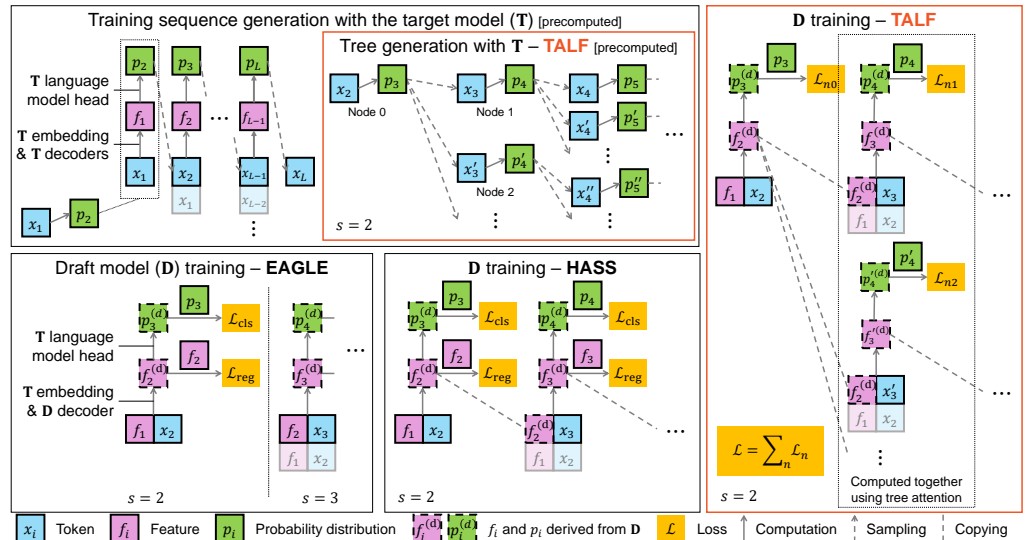

Figure 1: Draft model training process of EAGLE (Li et al., 2024b), HASS (Zhang et al., 2025), and TALF. With the precomputed training sequence $x_{1:L}$, training is performed for time steps $1 < s < L$.

During training, EAGLE aligns the draft model with the target LLM for features ($f_s^{(d)}$ with $f_s$) as well as the probability distributions ($p_{s+1}^{(d)}$ with $p_{s+1}$). First, the target LLM produces a training sequence $x_{1:L}$, their corresponding features ($f_{1:L-1}$), and probability distributions ($p_{1:L}$) through autoregressive iterations. Using them as soft labels, a drafting iteration ($p_{s+1}^{(d)}, f_s^{(d)} \leftarrow \mathbf{D}(x_{2:s}, f_{1:s-1})$), a loss calculation, and a gradient-based draft model update are performed for each time step $s$.

Despite numerous SpD proposals, most rely on standard knowledge distillation (Hinton et al., 2015) as their training objective, using the target LLM's output as soft labels. Similarly, EAGLE trains its draft model using a combination of two loss functions:

- Regression loss to align the features: $\mathcal{L}_{\text{reg}} = \|f_s - f_s^{(d)}\|_1$.

- Classification loss to align the probability distributions: $\mathcal{L}_{\text{cls}} = -\sum_v p_{s+1}(v) \cdot \log p_{s+1}^{(d)}(v)$.

HASS (Zhang et al., 2025) identifies a misalignment between training and inference in EAGLE. During inference, speculated features ($f_s^{(d)}, \cdots$) produced from the draft model are added to the inputs since the second drafting iteration. By contrast, EAGLE's drafting iteration during training only involves inputs produced by the target model ($x_{2:s}$ and $f_{1:s-1}$). To mitigate this, HASS modifies the training process and its loss function. The draft model follows through a short target LLM output sequence (e.g., $x_{1:s} \rightarrow x_{s+1} \rightarrow x_{s+2} \rightarrow p_{s+3}$), while feeding feature speculations generated by itself ($f_s^{(d)}$ and $f_{s+1}^{(d)}$) back as input (see Figure 1). Then, for this sequence, HASS gathers $\mathcal{L}_{\text{reg}}$ and $\mathcal{L}_{\text{cls}}$ losses comparing features and probability distributions generated by the draft model with those from the target LLM. In this way, HASS reflects the feature-related inference behavior of the draft model in the training objective. HASS also introduces a top-$K$ (e.g., $K = 10$) distillation loss, which can be orthogonally applied to $\mathcal{L}_{\text{cls}}$ to put more emphasis on $K$ highest-probability tokens.

## 2.3 TREE-BASED SPD & DYNAMIC TREE CONSTRUCTION: EAGLE-2 & SPECEXEC

Tree-based SpD (Miao et al., 2024) goes beyond verifying one sequence; it organizes multiple draft sequences into a tree structure and verifies the entire tree at once, effectively increasing the mean acceptance length ($\tau$). As a number of tokens can show high probabilities in $p_{s+1}^{(d)}$, we sample multiple tokens from $p_{s+1}^{(d)}$ and continue the next drafting iterations each with a different $x_{s+1}^{(d)}$. Repeating such will create a tree of candidate tokens, where each path from the root node ($x_s$) to a leaf node ($x_s$) represents a draft sequence. The entire tree can be verified by the target LLM in a single forward pass with little overhead by using tree attention (Miao et al., 2024).

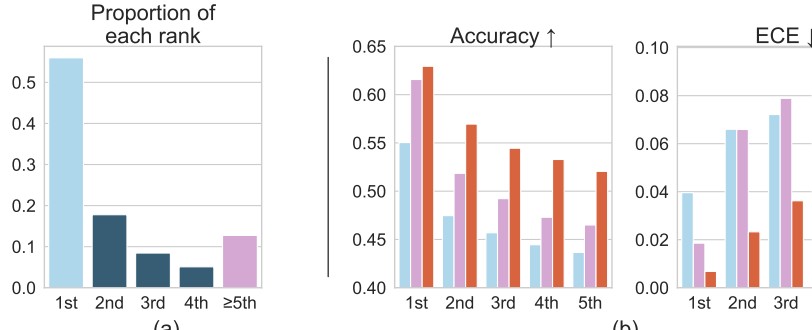

Figure 2: (a) Proportion of candidate tokens, which constitute the draft tree during inference, by rank on the probability distribution. (b) Top-1 accuracy and expected calibration error (ECE) for a draft model trained with EAGLE, HASS, and TALF when self-conditioned on a token ranked $n$-th ($n = 1, 2, 3, 4,$ and $5$) on the previous speculated probability distribution.

EAGLE-2 (Li et al., 2024a) constructs the tree dynamically during inference. It utilizes the speculated probability distribution ($p^{(d)}$) to compute $\Pr(x^{(d)} \mid x_{1:s})$ for each token (node) $x^{(d)}$ in the tree. EAGLE-2 performs a simple beam search (Ow & Morton, 1988) to find high-probability nodes with little drafting overhead. Meanwhile, the optimal tree construction approach from SpecExec (Svirschevski et al., 2024) has attempted to maximize the sum of the probabilities for a given number of nodes ($N$). However, this "optimal" approach involves a large search space, potentially damaging the end-to-end performance with increased drafting overhead. Refer to Appendix B for more details on prior dynamic tree construction methods.

## 3 METHOD

### 3.1 TRAINING-INFERENCE MISALIGNMENT FOR TREE-BASED SPD

We observe that, although inference relies on tree-based SpD, prior draft model training methods still use token sequences—rather than trees—generated by the target model as references. This creates a training-inference mismatch: the accuracy ($\tau$) of the draft model depends on how well its entire draft tree aligns with the target model, yet sequence-based training overlooks the opportunity to improve the draft model's ability to explore alternative tree nodes with fairly high probabilities.

For quantification, we evaluate how the draft models trained with EAGLE and HASS perform when self-conditioned on a lower-probability token. We used the setup described in §4.1 over a held-out test set $\mathcal{D}_{\text{test}}$, each element of which is an input sequence $x_{1:s}$. After the first drafting iteration producing $p_{s+1}^{(d)} \leftarrow \mathbf{D}(x_{2:s}, f_{1:s-1})$, we simulated inference by feeding the draft model's own prediction $x_{s+1}^{(d)}$ sampled from $p_{s+1}^{(d)}$ back as input to itself. We observed how the choice of $x_{s+1}^{(d)}$ in $p_{s+1}^{(d)}$ affects the next drafting iteration result ($p_{s+2}^{(d)}$), where $x_{s+1}^{(d)}$ is ranked $n$-th ($n = 1, 2, 3, 4, 5$) in $p_{s+1}^{(d)}$. Meanwhile, we also made the target LLM predict the ($s + 2$)-th token $\tilde{x}_{s+2}$ when conditioned on our $x_{s+1}^{(d)}$ choice. We measured how well $p_{s+1}^{(d)}$ predicts $\tilde{x}_{s+2}$ according to the choice of $x_{s+1}^{(d)}$ using accuracy and expected calibration error (ECE) (Guo et al., 2017) as metrics (detailed in Appendix A).

While HASS improves both ECE and accuracy when conditioned on 1st-ranked tokens, the gains are marginal or even negative for lower-ranked tokens (see Figure 2(b)), suggesting that draft models trained with HASS underperform on tree nodes outside the best draft sequence. Empirically, although 1st-ranked tokens possess the largest portion (roughly 55%) in the final draft tree, lower-ranked tokens are not negligible, where those ranked 5th or lower account for over 10% (see Figure 2(a)).

### 3.2 TRAINING WITH A TREE-AWARE LOSS FUNCTION (TALF)

These observations motivate us to propose a *tree-aware loss function (TALF)*, which aggregates cross-entropy loss terms over dynamic draft tree nodes, effectively reflecting the inference behavior of the

---

**Algorithm 1** TALF Training Procedure

---

**Require:** Target model $\mathbf{T}$, draft model $\mathbf{D}$, the number of nodes $N$
1: **for all** training sequence **do**
2:  $\mathbf{T}$ builds a tree $\mathcal{G}$ with $N$ nodes using a dynamic tree construction algorithm
3:  **for all** tree node $n$ in $\mathcal{G}$ **do**
4:    Let $x_{[n]}$ be the sequence of tokens represented by the node $n$
5:    $p_{\text{child}(n)} \leftarrow \mathbf{T}(x_{[n]})$          ▷ Precomputed during tree generation
6:    $p^{(d)}_{\text{child}(n)} \leftarrow \mathbf{D}(x_{[n]})$   ▷ Optimize with tree attention (input features omitted for brevity)
7:    Compute cross-entropy loss

$$\mathcal{L}_n = - \sum_{v \in \text{Vocab}} p_{\text{child}(n)}(v) \, \log p^{(d)}_{\text{child}(n)}(v)$$

8:    Backpropagate $\mathbf{D}$ with $\mathcal{L} = \sum_{n \in \mathcal{G}} \mathcal{L}_n$

---

draft model in the training objective. Compared to HASS, TALF yields only marginal improvements in the 1st-ranked case, but achieves 5% accuracy gains and 0.05 ECE drops for lower-ranked cases (see Figure 2(b)).

The training procedure with TALF is shown in Algorithm 1 and Figure 1. We first make the target model generate a tree dynamically, setting each token in a training sequence as the root node ($x_s \rightarrow p_{s+1}$ node). We make the draft model follow through the path from the root node to each node ($n$) of the tree, while feeding $f^{(d)}$ back as input as in HASS. We compute cross-entropy loss between the speculated ($p^{(d)}_{\text{child}(n)}$) and the target ($p_{\text{child}(n)}$) probability distributions for the token following $n$. Per-node losses are aggregated to compute the final loss.

Before training, the tree shape and the soft labels at each node (i.e., next-token predictions) are preprocessed by the target model. Making the draft model dynamically construct the tree at training time would generate a different tree structure for each training epoch, requiring multiple target model invocations. As this would incur prohibitively high computational cost, we make the target model fix the tree structure in advance, which can be reused for multiple training epochs.

As shown in Figure 1, we process multiple nodes on the same tree depth together using tree attention, which significantly accelerates the training process of TALF. We modify the attention-masking technique of HASS, which was originally designed for sequential inputs, to support tree structures. While any dynamic tree construction algorithms can be used, we employ the simple beam search method of EAGLE-2 for training. Sophisticated algorithms, such as the one used in SpecExec (see §2.3), would cause an increase in the preprocessing cost due to additional data structures that require complex handling (e.g., $\mathcal{Q}$ and $\mathcal{D}$ in Algorithm 2).

Unlike EAGLE and HASS, TALF does not use a regression loss for feature alignment. In our experiments, training solely on the token probability distributions across multiple nodes was sufficient for the model to learn to use features in an autoregressive manner, yielding better performance.

### 3.3 Stopping at Low Further Gains (SALF) in Dynamic Tree Construction

Existing dynamic tree construction methods either fail to include some high-probability tokens in the tree (e.g., beam search from EAGLE-2) or require a lot of drafting overhead to search for all possible high-probability tokens (e.g., optimal tree search from SpecExec). We develop Algorithm 2 to address these issues with a balanced dynamic tree construction method.

We extend the optimal tree search method by introducing *stopping at low further gains (SALF)*, a conditional criterion for efficient drafting. For constructing a tree with $N$ nodes, we use a priority queue $\mathcal{Q}$ with a fixed capacity of $N$. $\mathcal{Q}$ keeps high-probability nodes that have not yet been *expanded*, i.e., they have not been processed by the draft model to generate their child nodes. Per iteration, we pick $B$ highest-probability nodes from $\mathcal{Q}$ for expansion, move them to the output tree $\mathcal{G}$, and insert their child nodes into $\mathcal{Q}$. If we repeat this (Algorithm 2 without the red blocks), it is guaranteed to find the highest-probability nodes for the draft tree (see Appendix B for the proof).

---

**Algorithm 2** Dynamic Tree Construction with SALF

---

**Require:** Draft model $\mathbf{D}$, the number of nodes $N$, batch size $B$ ($\leq N$), SALF threshold $th \leq 1$

**Ensure:** Tree $\mathcal{G}$ containing the top-$N$ high-probability nodes

1: $\mathcal{G} \leftarrow \{(pr = 1, n = \text{root node})\}$          $\triangleright$ $\mathcal{G}$ contains up to $N$ nodes with high $pr$
2: Init a capacity-$N$ priority queue $\mathcal{Q} \leftarrow \mathcal{G}$    $\triangleright$ $\mathcal{Q}$ contains up to $N$ non-expanded nodes with high $pr$
3: **while** true **do**
4:      Init an empty list $\mathcal{D}$
5:      **for** b in 0..B **do**                              $\triangleright$ Select $B$ nodes to expand
6:          **if** $\mathcal{Q}$.empty() **then break**
7:          $(pr, n) \leftarrow \mathcal{Q}$.pop()
8:          $\mathcal{G}$.push($(pr, n)$)
9:          $\mathcal{D}$.append($(pr, n)$)
10:     $\epsilon \leftarrow \min\{pr \mid (pr, n) \in \mathcal{G}\}$
11:     **if** $|\mathcal{G}| < N$ **then** $\epsilon \leftarrow 0$
12:     $\mathcal{D} \leftarrow \{(pr, n) \in \mathcal{D} \mid pr > \epsilon\}$          $\triangleright$ Ignore nodes that cannot go into $\mathcal{G}$ due to low $pr$
13:     **if** $\sum_{(pr,n) \in \mathcal{D}} pr < th$ **then break**
14:     **for all** $(pr, n) \in \mathcal{D}$ **do**                $\triangleright$ Optimize with batched tree expansion
15:          Let $x_{[n]}$ be the sequence of tokens represented by the node $n$
16:          $p^{(d)}_{\text{child}(n)} \leftarrow \mathbf{D}(x_{[n]})$                 $\triangleright$ (input features omitted for brevity)
17:          **for all** $v \in$ Vocab **do**
18:             Let $n'$ be a child node of $n$ representing the next token $v$
19:             $pr' \leftarrow pr \cdot p^{(d)}_{\text{child}(n)}(v)$          $\triangleright$ Probability product calculation
20:             $\mathcal{Q}$.push($(pr', n')$)     $\triangleright$ $\mathcal{Q}$ automatically discards low-$pr$ entries when (# of entries) $> N$

---

However, the final goal of SpD is not to search for the highest-probability nodes but to minimize the end-to-end latency. We observe that the aforementioned process incurs excessive drafting overhead at deeper tree depths due to the large number of nodes with modest probabilities.

SALF (highlighted in red in Algorithm 2) addresses this problem by stopping when further drafting iterations is unlikely to increase the overall probability of the tokens included in the tree. Concretely, we stop drafting if the nodes in $\mathcal{D}$, whose entries are extracted from the top-$B$ highest-probability nodes in $\mathcal{Q}$, have a low probability sum. The probability sum monotonically decreases for each iteration (Theorem 1, see Appendix C for the proof), allowing us to precisely identify the point at which the expected benefit of further drafting falls below a configurable SALF threshold ($th$). SALF achieves significant speedups due to reduced drafting overhead.

**Theorem 1** (monotonically decreasing probability sum)**.** Let $\mathcal{D}_i$ denote the value of $\mathcal{D}$ at the early-stopping check (line 13) in the $i$-th iteration of the main loop in Algorithm 2. Define the sum of the probabilities of the entries in $\mathcal{D}_i$ as $S_i = \sum_{(pr,n) \in \mathcal{D}_i} pr$. Then, given that $B < |\text{Vocab}|$, the sequence $\{S_i\}$ is monotonically decreasing:

$$\forall i \geq 2, \quad S_i > S_{i+1}$$

## 4 EVALUATION

### 4.1 EXPERIMENTAL SETUP

**Models.** We used Llama-2-7B-Chat (Llama2-7B) (Touvron et al., 2023), Llama-3.1-8B-Instruct (Llama3-8B) (Llama Team, AI @ Meta, 2024), and DeepSeek-R1-Distill-Llama-8B (Deepseek-AI, 2025) for evaluation.

**Tasks.** We used test datasets with varying characteristics: MT-bench (Zheng et al., 2023), Humaneval (Chen et al., 2021), GSM8k (Cobbe et al., 2021), Alpaca (Taori et al., 2023), and CNN/Daily Mail (Nallapati et al., 2016). We set the inference batch size to one, following EAGLE and HASS.

Table 1: Mean SpD speedup (vs. execution without SpD) for various models, tasks, and temperatures. Relative mean improvements of SALF & TALF compared to EAGLE-2 and HASS are also shown.

| Model | Method | MT-bench | HumanEval | GSM8K | Alpaca | CNN/DM | Mean |
|-------|--------|----------|-----------|-------|--------|--------|------|
| **Temperature = 0** (Greedy) | | | | | | | |
| Llama2-7B | EAGLE-2 | 2.71× | 3.09× | 2.81× | 2.59× | 2.26× | 2.68× (+15.6%) |
| | HASS | 2.95× | 3.39× | 3.00× | 2.75× | 2.51× | 2.91× (+ 6.5%) |
| | SALF & TALF | 3.11× | 3.48× | 3.19× | 3.02× | 2.72× | 3.09× |
| Llama3-8B | EAGLE-2 | 2.10× | 2.44× | 2.21× | 2.11× | 1.82× | 2.13× (+35.0%) |
| | HASS | 2.33× | 2.89× | 2.58× | 2.35× | 2.05× | 2.42× (+18.4%) |
| | SALF & TALF | 2.79× | 3.25× | 2.99× | 2.81× | 2.56× | 2.87× |
| Deepseek-R1-Distill-Llama-8B | EAGLE-2 | 1.94× | 2.09× | 2.11× | 1.84× | 1.69× | 1.93× (+28.0%) |
| | HASS | 2.02× | 2.21× | 2.23× | 1.91× | 1.72× | 2.01× (+22.9%) |
| | SALF & TALF | 2.48× | 2.64× | 2.66× | 2.40× | 2.19× | 2.47× |
| **Temperature = 1** (Non-Greedy) | | | | | | | |
| Llama2-7B | EAGLE-2 | 2.52× | 2.94× | 2.72× | 2.42× | 2.20× | 2.55× (+18.0%) |
| | HASS | 2.78× | 3.10× | 2.96× | 2.76× | 2.36× | 2.78× (+ 8.1%) |
| | SALF & TALF | 2.96× | 3.25× | 3.19× | 3.00× | 2.67× | 3.01× |
| Llama3-8B | EAGLE-2 | 1.54× | 2.20× | 1.90× | 1.73× | 1.56× | 1.77× (+39.4%) |
| | HASS | 1.68× | 2.53× | 2.15× | 2.00× | 1.73× | 2.00× (+23.7%) |
| | SALF & TALF | 2.14× | 2.94× | 2.65× | 2.46× | 2.23× | 2.47× |
| Deepseek-R1-Distill-Llama-8B | EAGLE-2 | 1.70× | 1.94× | 2.01× | 1.70× | 1.58× | 1.78× (+28.4%) |
| | HASS | 1.79× | 2.02× | 2.09× | 1.74× | 1.59× | 1.84× (+24.4%) |
| | SALF & TALF | 2.22× | 2.46× | 2.54× | 2.21× | 2.03× | 2.28× |

**Training.** We trained the draft models with a ShareGPT (Aeala, 2023) dataset containing 68,000 dialogues. For Llama2-7B and Llama3-8B, we first trained the draft model for ten epochs using the original EAGLE loss. The trained draft model is used for evaluating EAGLE and EAGLE-2, which only differ in drafting. Then, we performed additional training with the ten-epoch-trained draft model using either HASS or TALF as a loss function for three epochs. For, Deepseek-R1-Distill-Llama-8B, we took a different approach to account for longer training time required for training with HASS or TALF. We trained each model (EAGLE, HASS, and TALF) for the same amount of time (24 hours on a system with two H100 80GB GPUs), which allows a fair comparison regarding the training cost.

**Hyperparameters.** For EAGLE and HASS, we set the classification loss weight and the top-10 distillation loss weight $10\times$ higher than the feature regression loss weight ($\lambda_{\text{cls}} = \lambda_{\text{distil}} = 10, \lambda_{\text{reg}} = 1$). We set the sequence/tree depth to three for HASS/TALF and used $k = 4$ to construct the tree when training with TALF. More training details are described in Appendix D.

**Inference.** We performed end-to-end LLM inference using the Hugging Face Transformers library with a PyTorch backend. For EAGLE-2 and HASS, we used their respective open-source implementations. We set $N = 60$, $k = 10$, and $depth = 7$ for beam search (used in EAGLE-2 and HASS). For SALF, we used Algorithm 2 with $N = 60$, $B = 10$, and a SALF threshold of $th = 0.6$ by default. All inference experiments were executed on a system with a single NVIDIA A100 80GB GPU (PCIe). Speedups relative to the baseline LLM inference without SpD were measured.

## 4.2 END-TO-END SPEEDUP

*SALF & TALF outperform existing methods across every model and dataset we tested.* Table 1 summarizes the end-to-end speedups (vs. execution without SpD) achieved by using SALF & TALF. Compared to EAGLE-2 and HASS, SALF & TALF achieve consistent mean improvements of 1.16–1.39× and 1.07–1.24×, respectively, under both greedy (temperature = 0) and non-greedy (temperature = 1) sampling cases. Furthermore, benefits of using SALF & TALF become more pronounced when stronger target LLMs (e.g., Deepseek-R1-Distill-Llama-8B) are employed. This stems from the greater difficulty draft models face in aligning with stronger target LLMs, a challenge that SALF & TALF address more effectively by enabling optimized tree-based SpD.

Table 2: Comparison of speedup and mean generation length ($\tau$) for various (tree construction method, loss) combinations. The target LLM is Deepseek-R1-Distill-Llama-8B.

| Tree constr. method | Loss | MT-bench | | HumanEval | | GSM8K | | Alpaca | | CNN/DM | | Mean | |
|---|---|---|---|---|---|---|---|---|---|---|---|---|---|
| | | Speedup | $\tau$ | Speedup | $\tau$ | Speedup | $\tau$ | Speedup | $\tau$ | Speedup | $\tau$ | Speedup | $\tau$ |
| Beam search | EAGLE-2 | 1.76× | 3.42 | 1.91× | 3.76 | 1.95× | 3.80 | 1.66× | 3.21 | 1.51× | 2.98 | 1.75× | 3.44 |
| | HASS | 1.84× | 3.61 | 2.03× | 3.98 | 2.09× | 4.04 | 1.75× | 3.39 | 1.55× | 3.06 | 1.84× | 3.62 |
| | TALF | 1.98× | 3.87 | 2.16× | 4.23 | 2.26× | 4.42 | 1.85× | 3.59 | 1.66× | 3.28 | 1.97× | 3.88 |
| Optimal tree search | EAGLE-2 | 1.94× | 3.54 | 2.09× | 3.87 | 2.11× | 3.88 | 1.84× | 3.36 | 1.69× | 3.13 | 1.93× | 3.56 |
| | HASS | 2.02× | 3.70 | 2.21× | 4.08 | 2.23× | 4.10 | 1.91× | 3.47 | 1.72× | 3.17 | 2.01× | 3.70 |
| | TALF | 2.19× | **3.98** | 2.34× | **4.30** | 2.44× | **4.49** | 2.04× | **3.71** | 1.84× | **3.40** | 2.16× | **3.98** |
| SALF | EAGLE-2 | 2.31× | 3.33 | 2.44× | 3.68 | 2.39× | 3.58 | 2.26× | 3.18 | 2.07× | 2.93 | 2.29× | 3.34 |
| | HASS | 2.41× | 3.60 | 2.55× | 3.99 | 2.52× | 3.96 | 2.30× | 3.39 | 2.12× | 3.09 | 2.37× | 3.61 |
| | TALF | **2.48×** | 3.73 | **2.64×** | 4.07 | **2.66×** | 4.16 | **2.40×** | 3.50 | **2.19×** | 3.20 | **2.47×** | 3.73 |

## 4.3 INDIVIDUAL BENEFITS OF SALF & TALF

We measured speedups and mean generation lengths ($\tau$) for different combinations of loss functions and tree-construction methods to isolate the benefits of SALF and TALF (see Table 2).

**TALF vs. prior loss functions.** When using the same tree construction method for all three loss functions, TALF improves $\tau$ over EAGLE-2 and HASS by 12.9/11.8/11.7% and 7.2/7.3/3.5% under beam/optimal/SALF tree search circumstances, respectively. These improvements are consistent across all benchmarks, showing that aligning the training objective with SpD drafting generalizes beyond any single task suite.

**SALF vs. prior tree construction methods.** For any loss function, moving from beam search to optimal tree search increases $\tau$. Per-benchmark deltas are consistently positive, indicating that the candidate token set from simple beam search is suboptimal, and that globally selecting the acceptance-maximizing tree provides additional gains beyond the loss function. When SALF is added to optimal tree search, $\tau$ decreases by 6.2%, 2.4%, and 6.3% for EAGLE-2, HASS, and TALF, respectively, while end-to-end performance increases by 18.6%, 17.9%, and 14.4%. As the model trained with TALF is better calibrated on lower-ranked branches, it has fewer wasteful nodes that are ignored with SALF, yielding smaller incremental speedups than the cases with EAGLE-2 or HASS.

## 4.4 PARAMETER SENSITIVITY

Table 3: Mean generation length ($\tau$) with different top-$k$ settings for training TALF. The tree size used for training increases for larger $k$. The target LLM is Deepseek-R1-Distill-Llama-8B.

| Loss (top-$k$) | MT-bench | HumanEval | GSM8K |
|---|---|---|---|
| HASS (top-1) | 3.70 | 4.08 | 4.10 |
| TALF (top-1) | 3.71 | 4.08 | 4.31 |
| TALF (top-2) | 3.83 | 4.23 | 4.48 |
| TALF (top-4) | **3.98** | **4.30** | **4.49** |

**Top-$k$ for training.** Table 3 reports $\tau$ on three benchmarks as we vary $k$, which decides how wide we will look to construct the tree during training; up to $k$-th-ranked tokens will be included in the tree. TALF with $k = 1$ is almost the same as HASS. As we increase $k$, more lower-ranked tokens are considered, enhancing $\tau$. We chose to use $k = 4$ as our default setting to obtain higher $\tau$.

**SALF threshold ($th$) for drafting.** We measured the impact of using different SALF thresholds ($th$) across three benchmarks. For lower $th$, we get closer to generating a draft tree with highest-probability tokens, leading to higher $\tau$. However, when $th$ is too low, the actual end-to-end latency may increase due to excessive drafting overhead. Using $th = 0.5$ yields the highest mean speedup

Table 4: Performance with various SALF thresholds ($th$) used for tree construction during drafting. The target LLM is Deepseek-R1-Distill-Llama-8B.

| Threshold ($th$) | MT-bench | | HumanEval | | GSM8K | | Mean | |
|---|---|---|---|---|---|---|---|---|
| | Speedup | $\tau$ | Speedup | $\tau$ | Speedup | $\tau$ | Speedup | $\tau$ |
| 0.0 | 2.19× | 3.98 | 2.34× | 4.30 | 2.44× | 4.49 | 2.32× | 4.26 |
| 0.1 | 2.25× | 3.99 | 2.36× | 4.31 | 2.46× | 4.49 | 2.36× | 4.26 |
| 0.2 | 2.37× | 3.97 | 2.46× | 4.31 | 2.56× | 4.49 | 2.46× | 4.26 |
| 0.3 | 2.45× | 3.94 | 2.58× | 4.30 | 2.63× | 4.45 | 2.55× | 4.23 |
| 0.4 | 2.49× | 3.90 | 2.62× | 4.26 | 2.64× | 4.38 | 2.58× | 4.18 |
| 0.5 | **2.52×** | 3.82 | **2.65×** | 4.17 | **2.69×** | 4.29 | **2.62×** | 4.10 |
| 0.6 | 2.48× | 3.73 | 2.64× | 4.07 | 2.66× | 4.16 | 2.59× | 3.99 |
| 0.7 | 2.47× | 3.59 | 2.63× | 3.92 | 2.65× | 3.99 | 2.58× | 3.83 |
| 0.8 | 2.39× | 3.38 | 2.54× | 3.72 | 2.59× | 3.78 | 2.51× | 3.63 |
| 0.9 | 2.27× | 3.11 | 2.43× | 3.40 | 2.43× | 3.44 | 2.37× | 3.32 |

for Deepseek-R1-Distill-Llam-8B (2.62×) based on Table 4. However, we observed more consistent performance improvements for the tested target LLMs when $th = 0.6$ (default). Tuning $th$ based on the model or adapting it dynamically during inference is a potential direction for future work.

## 5    RELATED WORK

While the draft-verify paradigm of SpD can be attributed to blockwise decoding (Stern et al., 2018), Leviathan et al. (2023) and Chen et al. (2023) developed a verification method based on rejection sampling, which allows an exact simulation of the target LLM's output probability distribution. Tree-based SpD, first proposed in SpecInfer (Miao et al., 2024), exploits tree attention to verify a draft tree in a single forward pass. Tree attention adds a tree attention mask, which represents the causality (edges) within the tree, to a regular attention (Vaswani et al., 2017) layer. Numerous following studies (Zhang et al., 2025; Hu et al., 2025; Li et al., 2024b;a; Miao et al., 2024; Chen et al., 2024a; Spector & Re, 2023; Sun et al., 2023; Ankner et al., 2024; Cai et al., 2024; Svirschevski et al., 2024) have adopted tree-based SpD. While early work has used a fixed (e.g., Sequoia (Chen et al., 2024a)) tree structure, recent studies (Li et al., 2024a; Svirschevski et al., 2024; Zhang et al., 2025; Hu et al., 2025) dynamically construct the tree during drafting. Further improvements, such as dynamically adjusting the tree depth in AdaEagle (Zhang et al., 2024b), have also been proposed.

Also, various approaches have been proposed for drafting, such as using a smaller model from the same LLM family (Sun et al., 2023; Chen et al., 2024b; Spector & Re, 2023; Leviathan et al., 2023; Chen et al., 2023; He et al., 2024), employing an independent fine-tuned model (Kim et al., 2023; Miao et al., 2024; Liu et al., 2024b; Zhou et al., 2024), inserting additional heads for drafting into the target LLM (Cai et al., 2024; Ankner et al., 2024; Stern et al., 2018), and reusing the target LLM while skipping some of its layers during drafting (Zhang et al., 2024a; Elhoushi et al., 2024; Liu et al., 2024a; Xia et al., 2025). EAGLE (Li et al., 2024b) is a hybrid approach employing a fine-tuned draft model that also leverages features from the target LLM, extending the additional-head approach.

## 6    CONCLUSION

We have identified a fundamental mismatch between how existing tree-based SpD methods are trained and how they are used at inference. To address this, we introduced TALF, a novel loss function and its training procedure that aligns the draft model's output distribution over the entire expansion tree with that of the target LLM. We further developed a systematic approach for building the tree dynamically through SALF with a provable monotonicity guarantee, reducing unnecessary drafting computations during inference. Experiments on various benchmarks show that the combined use of SALF & TALF consistently outperforms state-of-the-art SpD solutions, EAGLE-2 and HASS, yielding 1.16–1.39× and 1.07–1.24× wall-clock speedups without any generation quality degradation.

## REPRODUCIBILITY STATEMENT

All code used for data preprocessing, model training, evaluation, and figure generation is provided in the supplementary materials, including scripts such as preprocess.py, train.py, evaluate.py, calibration.py, and graph.py. Detailed descriptions of the experimental setup, model architectures, and hyperparameters are included in §4.1 and §D. The datasets used in our experiments are either publicly available or described in detail, with data processing steps documented in the supplementary materials. For theoretical results, all assumptions and complete proofs are provided in the §B and §C.

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

## A  METRICS USED IN §3.1

For the experimental setup detailed in §3.1, we use the following two metrics to assess how well $p_{s+2}^{(d)}$ predicts $\tilde{x}_{s+2}$ for a test set $\mathcal{D}_{\text{test}}$:

- **Top-1 accuracy** (higher is better):

$$\text{Acc}(\mathcal{D}_{\text{test}}) = \frac{1}{|\mathcal{D}_{\text{test}}|} \sum_{x_{1:s} \in \mathcal{D}_{\text{test}}} \mathbf{1}\big\{ \arg \max_{v \in \text{Vocab}} \big(p_{s+2}^{(d)}(v)\big) = \tilde{x}_{s+2} \big\}$$

- **Expected calibration error (ECE)** (lower is better) quantitatively measures statistical calibration by comparing accuracy and confidence (Guo et al., 2017):

$$\text{ECE} = \sum_{m=1}^{M} \frac{|\mathcal{B}_m|}{|\mathcal{D}_{\text{test}}|} \big|\text{Acc}(\mathcal{B}_m) - \text{Conf}(\mathcal{B}_m)\big|,$$

where $\mathcal{D}_{\text{test}}$ is binned into $M$ equisized bins $\mathcal{B}_m$ by the confidence ($p_{s+2}^{(d)}(\tilde{x}_{s+2})$) value and

$$\text{Conf}(\mathcal{B}_m) = \frac{1}{|\mathcal{B}_m|} \sum_{x_{1:s} \in \mathcal{B}_m} p_{s+2}^{(d)}(\tilde{x}_{s+2}).$$

## B  DYNAMIC TREE CONSTRUCTION ALGORITHMS

Dynamic tree construction algorithms in speculative decoding aim to efficiently select high-probability tokens by dynamically expanding a tree structure based on probability products. EAGLE-2 (Li et al., 2024a) employs a dynamic tree construction method that expands only the top-$k$ nodes at the deepest tree depth according to their global acceptance probabilities $V_i$. Concretely, the global acceptance probability $V_n$ for a node $n$ is computed as the product of the acceptance probabilities along the path from the root node $n_{\text{root}}$ to $n$:

$$V_n = \prod_{n' \in \text{path}(n_{\text{root}}, n) \setminus \{n_{\text{root}}\}} \Pr(n' \mid \text{parent}(n')) \approx \prod_{n' \in \text{path}(n_{\text{root}}, n) \setminus \{n_{\text{root}}\}} p_{\text{child}(\text{parent}(n'))}^{(d)}(n').$$

The true probability distributions are unknown at inference time and, thus, are approximated by the probability distributions produced by the draft model.

### BEAM SEARCH

The tree search algorithm (illustrated in Figure 4(a)) of EAGLE-2 Li et al. (2024a) is based on a simple beam search and consists of two phases:

1. **Tree expansion** ($k$ and $depth$ are parameters): At each expansion step, top-$k$ nodes from the deepest tree depth are selected. Each node is expanded by processing with the draft model to generate up to $k$ child nodes belonging to the next depth. This iterative expansion process continues until the tree reaches a predefined $depth$.

2. **Re-ranking** ($N$ is a parameter): After tree expansion, retain only *top-N* tokens with the highest $V_n$ values in the tree for verification.

However, this strategy is suboptimal; as it selects the next nodes to expand only in the deepest tree depth, it may overlook higher-probability nodes elsewhere in the tree. For an exemplar tree in Figure 3, a truly optimal tree expansion would prioritize the yellow nodes with higher $V_n$ over the pink nodes, which would be selected by EAGLE-2.

### OPTIMAL TREE SEARCH

Our algorithm builds upon the optimal tree search method (Svirschevski et al., 2024). Specifically, we provide the formulation of the optimal tree search problem as follows:

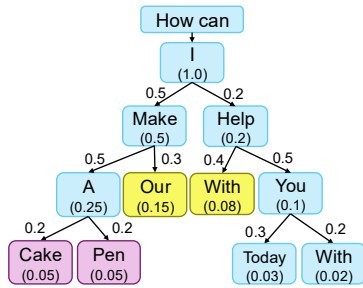

Figure 3: Tree expansion process with EAGLE-2 ($k = 2$). Numbers in parentheses indicate global acceptance probabilities ($V_n$) approximated by using the draft model.

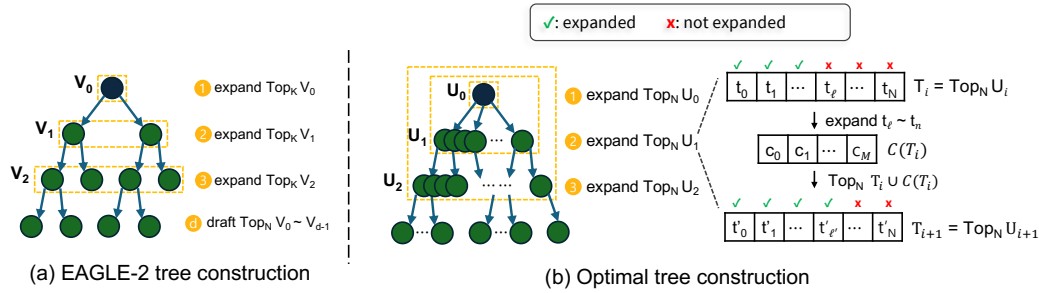

(a) EAGLE-2 tree construction    (b) Optimal tree construction

Figure 4: Tree expansion strategies: EAGLE-2 expands from deepest nodes, while optimal tree construction expands across all depth.

**Definition B.1** (Optimal tree $\mathcal{G}_{\mathrm{opt}}$). Given maximum tree depth $d$ and size $N$, the optimal tree $\mathcal{G}_{\mathrm{opt}}$ consists of exactly $N$ nodes, each nodes with depth$\leq d$, maximizing the sum of global acceptance probabilities.

**Definition B.2** ($\mathrm{Top}_N$ operator). Let $\mathcal{G}$ be a rooted tree with nodes $n$ having a value $v(n)$ and depth $\mathrm{depth}(n)$ (number of edges from the root). Define subset $U_i = \{\, n \in \mathcal{G} \mid \mathrm{depth}(n) \leq i \,\}$. For a set $X \subseteq \mathcal{G}$, $\mathrm{Top}_N(X)$ selects the N nodes with highest value:

$$\mathrm{Top}_N(X) := \arg\max_{\substack{S \subseteq X \\ |S| = N}} \sum_{n \in S} \mathrm{v}(n).$$

**Definition B.3** (Monotonic Tree). A tree $\mathcal{G}$ is *monotonic* if each node $n \in \mathcal{G}$ satisfies

$$\forall c \in \mathrm{child}(n), \quad v(c) < v(n),$$

where $\mathrm{child}(n)$ denotes the child nodes of $n$.

Definition B.1 imposes two critical constraints on the search algorithm: the maximal tree depth and the fixed number of candidate nodes. First, the depth constraint reflects the cost of draft model inference and also the diminishing returns associated with deeper searches; intuitively, the potential maximum gain decreases as the depth $i$ increases, bounded by $N - i$. Second, due to the computational expense of target model inference, it is crucial to batch only a limited number of candidate tokens efficiently.

Figure 4(b) illustrates the optimal tree search process employed in algorithm 2. In summary, at iteration $i$, the algorithm seeks to expand $\mathrm{Top}_N(U_i)$. Due to exponential growth of $U_i$, with increasing depth $i$, directly managing the set is impractical. Thus, the algorithm maintains a fixed size priority queue $T_i$ with capacity $N$, updating it iteratively according to

$$T_{i+1} = \mathrm{Top}_N(T_i \cup \mathrm{child}(T_i)).$$

We now present Theorems 2 and 3, establishing the correctness of Algorithm 2 (without early stopping) in identifying the optimal tree.

**Theorem 2** (Loop Invariant). For every $i \geq 0$,

$$T_i = \mathrm{Top}_N(U_i).$$

*Proof.* We proceed by induction on $i$.

*Base case ($i = 0$).* Since $U_0$ contains only the root node, it trivially follows that: $T_0 = \mathrm{Top}_N(U_0)$.

*Inductive step.* Assume $T_k = \mathrm{Top}_N(U_k)$ holds for some $k \geq 0$. Note that:

$$U_{k+1} = U_k \cup V_{k+1}, \quad V_{k+1} = \{n \mid depth(n) = k+1\} = \bigcup_{p \in V_k} \mathrm{child}(p).$$

Because of monotonicity, any node $p \in U_k \setminus T_k$ satisfies

$$v(p) \leq \min_{n \in T_k} v(n) \leq \min_{n \in T_{k+1}} v(n).$$

Hence, all its children $c \in \mathrm{child}(p)$ have lower values and cannot enter $\mathrm{Top}_N(U_{k+1})$. Therefore, the only valid new candidates at depth $k+1$ are children of nodes in $T_k$, i.e. $C_{k+1}$. Thus,

$$\mathrm{Top}_N(U_{k+1}) = \mathrm{Top}_N(T_k \cup C_{k+1}) = T_{k+1},$$

completing the induction. $\square$

**Theorem 3** (Termination). When the algorithm halts, the final set $T$ equals $\mathrm{Top}_N(\mathcal{T})$, the $N$ highest-value nodes in the entire tree.

*Proof.* Let

$$d = \max\{depth(n) \mid n \in \mathrm{Top}_N(\mathcal{T})\}.$$

From Theorem 2, after $d$ iterations,

$$T_d = \mathrm{Top}_N(U_d).$$

Since all node of depth $\leq d$ lies in $U_d$, and no deeper node can surpass their ancestors due to monotonicity, it follows that $\mathrm{Top}_N(U_d) = \mathrm{Top}_N(\mathcal{T})$. No further changes occur beyond depth $d$, thus the algorithm halts with: $T_{\mathrm{final}} = T_d = \mathrm{Top}_N(\mathcal{T})$. $\square$

## C  PROOF OF THEOREM 1

Before beginning the proof, if $B < |\mathrm{Vocab}|$, we can observe that $\mathcal{Q}$ always has $\geq B$ elements after the first iteration; when at least a single node was expanded in the previous iteration, each expanded node will try to push $|\mathrm{Vocab}| > B$ child nodes into $\mathcal{Q}$. In a practical implementation, as $\mathcal{Q}$ will only maintain $N (\geq B)$ elements, we do not need to push all $|\mathrm{Vocab}|$ childs but only a few high-probability childs. Such an implementation detail does not affect the correctness of Algorithm 2 or Theorem 1 as it will still produce the same $\mathcal{Q}$.

*Proof.* Partition the sets as follows:

$$\mathcal{D}_i = X_i \cup Y_i, \ X_i \cap Y_i = \emptyset, \quad \mathcal{D}_{i+1} = Z_{i+1} \cup W_{i+1}, \ Z_{i+1} \cap W_{i+1} = \emptyset,$$

where

$$X_i = \big\{(pr, n) \in \mathcal{D}_i \mid \exists\, (pr', n') \in \mathcal{D}_{i+1} : \mathrm{parent}(n') = n\big\}, \quad Y_i = \mathcal{D}_i \setminus X_i.$$

$$Z_{i+1} = \big\{(pr', n') \in \mathcal{D}_{i+1} \mid \mathrm{parent}(n') \in \mathcal{D}_i\big\}, \quad W_{i+1} = \mathcal{D}_{i+1} \setminus Z_{i+1},$$

For any $(pr', n') \in Z_{i+1}$, $n'$ is a child of only one parent $n$, where $(pr, n) \in X_i$. Then, for

$$Z_{i+1,n} = \{(pr', n') \in Z_{i+1} \mid \mathrm{parent}(n') = n\},$$

$\{Z_{i+1,n}\}_{(n,pr)\in X_i}$ is a partition of $Z_{i+1}$. As $|Z_{i+1,n}| \leq |\mathcal{D}_{i+1}| \leq B < |\text{Vocab}| = |\text{child}(n)|$,

$$
\begin{aligned}
\sum_{(pr',n')\in Z_{i+1}} pr' &= \sum_{(pr,n)\in X_i} \sum_{(pr',n')\in Z_{i+1,n}} pr' \\
&= \sum_{(pr,n)\in X_i} \sum_{(pr',n')\in Z_{i+1,n}} pr \cdot p^{(d)}_{\text{child}(n)}(n') \quad \text{(probability product)} \\
&= \sum_{(pr,n)\in X_i} pr \cdot \sum_{(pr',n')\in Z_{i+1,n}} p^{(d)}_{\text{child}(n)}(n') \\
&< \sum_{(pr,n)\in X_i} pr \cdot \sum_{n'\in\text{child}(n)} p^{(d)}_{\text{child}(n)}(n') \\
&= \sum_{(pr,n)\in X_i} pr \quad \text{(total probability)}
\end{aligned}
$$

Thus,

$$
\sum_{(pr,n)\in\mathcal{D}_i} pr > \sum_{(pr',n')\in Z_{i+1}} pr' + \sum_{(pr,n)\in Y_i} pr.
$$

Also, an element $(pr',n') \in W_{i+1}$ is an element of $Q$ not chosen to be included in $\mathcal{D}_i$ from the previous iteration due to its lower priority $(pr')$. Therefore, for any $(pr,n) \in Y_i \subset \mathcal{D}_i$, $pr \geq pr'$; i.e.,

$$
\forall (pr,n) \in Y_i, \quad pr \geq \max_{(pr',n')\in W_{i+1}} pr'.
$$

Now, we want to show that $|W_{i+1}| \leq |Y_i|$. Note that $|Z_{i+1}| \geq |X_i|$ because an element in $|X_i|$ has one or more non-overlapping child in $|Z_{i+1}|$. When $|\mathcal{D}_i| = B$, $|W_{i+1}| = |\mathcal{D}_{i+1}| - |Z_{i+1}| \leq |\mathcal{D}_i| - |X_i| = |Y_i|$ holds.

When $|\mathcal{D}_i| < B$, this means that some of the $B$ elements chosen from $\mathcal{Q}_i$ were smaller than $\epsilon_i$. Then, because $W_{i+1}$ consists of elements of $\mathcal{Q}_{i+1}$ that has not been expanded in the previous iteration, they must come from the leftovers of $\mathcal{Q}_i$, which are smaller than $\epsilon_i$. Also, because $\mathcal{G}$ only retains high-probability nodes, $\epsilon_{i+1} \geq \epsilon_i$ holds. Therefore, $pr' < \epsilon_{i+1}$ for any $(pr',n') \in W_{i+1}$, which means that they must have been discarded in Line 12 of Algorithm 2. This leads to $W_{i+1} = \emptyset$ and $|W_{i+1}| \leq |Y_i|$ trivially holds.

Putting these all together,

$$
\begin{aligned}
\sum_{(pr,n)\in\mathcal{D}_i} pr &> \sum_{(pr',n')\in Z_{i+1}} pr' + \sum_{(pr,n)\in Y_i} pr \\
&\geq \sum_{(pr',n')\in Z_{i+1}} pr' + |Y_i| \max_{(pr',n')\in W_{i+1}} pr' \\
&\geq \sum_{(pr',n')\in Z_{i+1}} pr' + |W_{i+1}| \max_{(pr',n')\in W_{i+1}} pr' \\
&\geq \sum_{(pr',n')\in Z_{i+1}} pr' + \sum_{(pr',n')\in W_{i+1}} pr' \\
&= \sum_{(pr',n')\in\mathcal{D}_{i+1}} pr'
\end{aligned}
$$

Hence, $S_i > S_{i+1}$, proving the monotonic decreasing property of $\{S_i\}$. $\qquad\square$

## D EXPERIMENTAL DETAILS AND ASSETS USED

We used the same ShareGPT dataset for all training runs, allocating 95 % of the data to training and 5 % to validation. The maximum sequence length was set to 2,048 tokens, and the tree (or sequence) depth during training was fixed to three. As in EAGLE and HASS, we built questions

and answers from a fixed prompt template and trained the draft model to predict the target model's outputs. We set the learning rate to 3e-4 and employed a cosine-annealing scheduler with a warm-up phase. The entire training procedure was designed to finish within one day on two NVIDIA A100 80 GB GPUs. We used the AdamW optimizer with beta = (0.9, 0.95) (Loshchilov & Hutter, 2019). The core implementation relies on PyTorch and Hugging Face's Transformers library, while the optimal-tree construction with SALF logic was written in C++ (using OpenMP and the PyTorch C++ API) and was bound to Python via pybind11.

Inference for each method was performed with the following setting:

- Vanilla (without SpD): We used the Huggingface Transformers library with a PyTorch backend.
- Beam search: We set the number of total candidate tokens in a tree ($N$) to 60, the top-$k$ to 10, and the tree depth to 7.
- SALF: We set the number of total candidate tokens in a tree ($N$) to 60, batch size ($B$) to 10, and the SALF threshold ($th$) to 0.6. Using a SALF threshold of 0.0 is equivalent to the optimal tree search method.

## E  DECLARATION OF LLM USAGE

The authors of this manuscript declare that LLMs were used only for writing, editing, or formatting purposes and that the LLM usage does not impact the core methodology, scientific rigorousness, or originality of the research.

