# OpenReview forum: "SALF & TALF: Optimized Loss Function and Drafting for Tree-based Speculative Decoding"
_ICLR.cc/2026/Conference — Submitted to ICLR 2026_

### Official Review · Reviewer_6X3v · 2025-10-16

**Soundness:** 3
**Presentation:** 3
**Contribution:** 2
**Rating:** 4
**Confidence:** 5

**Summary:**

This paper focuses on proposing a tree-aware loss function that explicitly incorporates the tree structure into draft model training to improve the speedup of current SPD. TALF aligns the draft model’s predictions with the target across all branches, mitigating the misalignment. They also improve the tree construction process in drafting with stopping at low further gains. The results show that they can deliver speedup over Eagle2 and HASS.

**Strengths:**

This paper is technically sound and easy to understand.

The experimental results show the effectiveness of the proposed method.

**Weaknesses:**

The paper lacks comparison to state-of-the-art methods such as Eagle3.

The paper focuses on generating tree structure and improve the overall MAT to speedup the large language model. However, they only conduct experiments with HuggingFace Transformers framework. Here comes a problem that the method may not have such speedup on the popular inference framework such as vLLM. In fact, HuggingFace Transformers framework does not optimize the speed of LLMs very well, which makes the ratio of the latency of tree generation process smaller. When using vLLM framework where the operations in LLMs are optimized very well, the tree generation process will take more time and reduce the speedup. The author should verify their method on such inference frameworks to show that their method is actually useful in reality.

**Questions:**

See weaknesses above.

---

> ### Author Response · Authors · 2025-11-26
> **Response by Authors to 6X3v (1/2)**
>
> We would like to express our sincere thanks for your considerate review and helpful feedback.
>
> 1. Comparison with EAGLE-3
>
>  We agree that comparison against EAGLE-3 is important for positioning our contribution. To address this, we trained an EAGLE-3 draft model with TALF loss using the ShareGPT dataset and observed that TALF continues to improve performance even in this setting.
>
> | Drafter variant              | MT-bench | HumanEval | GSM8k |
> |-----------------------------|:-------------:|:-------------:|:-------------:|
> | EAGLE-3, with feature loss  |        4.63  | 5.03 | 5.44 |
> | EAGLE-3, without feature loss    |        5.09  | 5.55 | 6.00 |
> | EAGLE-3, TALF               |        5.37  | 5.80 |  6.25 |
>
>  These results indicate that TALF is complementary to the architectural improvements of EAGLE-3 and that our method remains beneficial compared to the loss function employed in EAGLE-3.
>
> 2. Evaluation on vLLM and realistic serving frameworks
>
>  We fully share the concern that results based only on the HuggingFace Transformers framework may not reflect the behavior of modern highly optimized serving frameworks like vLLM. Many speculative decoding methods look strong in theory yet are difficult to adopt in practice. To directly address this gap, we implemented a tree-based speculative decoding system with batching on top of vLLM, which we refer to as vLLM-SpD. This framework extends vLLM with tree-based drafting and uses a Triton implementation of FlashAttention that supports tree masking. Because this Triton kernel is not as optimized as hand-tuned C++ CUDA kernels, our current implementation is conservative with respect to the possible speedups and could be further improved by further kernel optimizations.
>  Even under this conservative setup, we find that it is easy to identify SALF hyperparameters that outperform EAGLE-style trees and non-tree speculative decoding in vLLM. For example, on DeepSeek-R1-Distill-Llama-8B, vLLM-SpD with EAGLE-based speculative decoding runs $3.12\times$ faster than latest vLLM (v0.11.0) without speculative decoding and $2.02\times$ faster than vLLM with sequential speculative decoding. Within the same vLLM-SpD framework, SALF with a simple grid search over thresholds achieves even higher performance compared to EAGLE trees, both for batch size one and for batched inference. These results support that our method remains useful when integrated into a realistic, optimized serving system rather than only into a vanilla Transformers implementation.
> All experiments are conducted on a single NVIDIA A100 GPU. We constructed a heterogeneous workload by mixing MT-bench, HumanEval, GSM8K, Alpaca, and CNN/DM dataset, and issue requests to the serving framework at a rate of 0.8 requests per second while measuring end-to-end (E2E) latency from arrival to completion. For both EAGLE and SALF, we tuned the hyperparameters (top-$k$, depth, and $N$) to their best-performing values under this experimental setup.
> | Method | top-$k$ | Depth | $N$ | Threshold | Mean E2E (s) | p90 E2E (s) | p95 E2E (s) | p99 E2E (s) |
> |:------:|:-------:|:-----:|:---:|:---------:|:-------------:|:------------:|:------------:|:------------:|
> | EAGLE  |   16    |   4   | 60  |     --    |     11.5      |     20.2     |     23.8     |     27.1     |
> | SALF   |   16    |  10   | 60  |   0.0     |     14.9      |     26.1     |     30.5     |     36.5     |
> | SALF   |   16    |  10   | 60  |   0.1     |     12.1      |     20.6     |     24.3     |     30.1     |
> | SALF   |   16    |  10   | 60  |   0.2     |     10.4      |     17.7     |     20.3     |     23.9     |
> | SALF   |   16    |  10   | 60  |   0.3     |     10.2      |     17.1     |     19.8     |     22.5     |
> | SALF   |   16    |  10   | 60  |   0.4     |      **9.72 (-15\%)** |     **16.1 (-20 \%)**    |     **18.1 (-24 \%)**    |     **21.8 (-20 \%)**  |
> | SALF   |   16    |  10   | 60  |   0.5     |     10.0      |     16.6     |     18.3     |     21.7     |
> | SALF   |   16    |  10   | 60  |   0.6     |     10.0      |     17.0     |     18.8     |     22.8     |
> | SALF   |   16    |  10   | 60  |   0.7     |     10.7      |     18.4     |     20.5     |     22.1     |
> | SALF   |   16    |  10   | 60  |   0.8     |     11.1      |     18.6     |     20.6     |     23.9     |
> | SALF   |   16    |  10   | 60  |   0.9     |     12.1      |     20.2     |     22.6     |     25.6     |
>
> The results in the table isolate the impact of the tree construction algorithms in EAGLE and SALF and are therefore independent of TALF. When TALF is additionally applied, the performance gap between baseline and SALF & TALF becomes even larger.

---

> ### Author Response · Authors · 2025-11-26
> **Response by Authors to 6X3v (2/2)**
>
> 3. Broader deployment scenarios and metrics beyond a single framework
>
>  We agree that performance on vLLM is important, but we also want to emphasize that speculative decoding is deployed in a wide range of hardware and software environments. In consumer GPUs or mobile LLM deployments, the target model is much slower and the relative cost of tree generation becomes smaller, which makes methods that reduce target model computation even more attractive. In addition, as real systems increasingly prioritize energy per token, eliminating unnecessary drafting with SALF and achieving longer acceptance lengths with TALF makes the SALF & TALF particularly valuable in such scenarios.

---

### Official Review · Reviewer_QFQX · 2025-10-22

**Soundness:** 3
**Presentation:** 3
**Contribution:** 3
**Rating:** 6
**Confidence:** 3

**Summary:**

This paper presents a strong contribution to speculative decoding by addressing a clear training-inference mismatch in tree-based methods. The proposed TALF and SALF are novel, well-motivated, and demonstrate significant empirical improvements over state-of-the-art baselines. The work is timely, well-executed, and merits acceptance.

**Strengths:**

1. Novel and Well-Motivated Problem Formulation:​​ The paper convincingly identifies a critical yet overlooked issue: the misalignment between sequence-based training and tree-based inference in speculative decoding. The motivation is powerfully supported by empirical evidence showing the poor calibration of existing draft models on lower-ranked tokens.
2. ​Effective and Orthogonal Solutions:​​ The two contributions, TALF (training) and SALF (inference), address distinct parts of the pipeline and are shown to be complementary.
3. ​Extensive and Convincing Empirical Validation:​​ The experiments are thorough, evaluating multiple models (Llama2-7B, Llama3-8B, DeepSeek-R1), diverse tasks, and different sampling temperatures.

**Weaknesses:**

1. Inadequate Justification for Omitting Feature Loss:​​ The decision to remove the feature regression loss in TALF, a key component of EAGLE and HASS designed to prevent feature drift, is not sufficiently justified.

2. Limited Discussion on TALF Precomputation and Generalization:​​ The paper lacks details on the computational cost of precomputing draft trees with the target LLM for TALF training. Furthermore, it should be discussed whether fixing the tree structures from a static dataset (ShareGPT) might limit the draft model's ability to generalize to unseen prompt distributions or dynamic tree-building strategies during inference.

3. Ambiguity in SALF Threshold Interpretation:​​ While the SALF threshold is shown to be effective, its intuitive meaning is somewhat ambiguous. A more detailed interpretation of what the threshold value represents in terms of expected probability gain would aid in understanding and practical tuning.

**Questions:**

1. TALF Generalization:​​ The trees used for TALF training are precomputed on a specific dataset. How does the performance of a TALF-trained model generalize to prompts or domains significantly different from its training data? Is there a risk of overfitting to the specific tree structures generated from ShareGPT?

2. ​Feature Alignment Evidence:​​ Can the authors provide quantitative evidence to demonstrate that a TALF-trained model maintains feature alignment with the target LLM despite the removal of the explicit feature regression loss ?

3. Adaptive SALF Threshold:​​ The SALF threshold is a fixed hyperparameter. Have the authors explored making it adaptive based on runtime statistics to dynamically balance quality and overhead across different stages of generation or tasks?

---

> ### Author Response · Authors · 2025-11-26
> **Response by Authors to QFQX**
>
> We are grateful for your careful review and the insightful comments you provided.
>
> 1. Feature loss and feature alignment
>
> Our tree-based supervision guides the draft model purely at the token level, focusing on the probability of acceptance at each node in the speculative tree. This supervision aligns directly with the behavior of tree-based decoding, where runtime decisions rely solely on token probabilities and acceptance checks. In contrast, the feature regression loss optimizes hidden-state similarity to the target model, which is not directly used by the decoding algorithm. While this objective may perform reasonably in controlled settings, it introduces a mismatch with the true decoding objective and can divert the model's learning capacity away from behaviors that improve acceptance.
>
> During our initial experiments to validate this rationale, we observed that removing the feature regression loss degrades speculative decoding performance. In HASS, ablating the feature-loss term led to higher acceptance lengths and better end-to-end speedups, particularly under tree-based decoding. Moreover, TALF is defined purely over token probabilities on tree nodes, making it difficult to incorporate feature regression in a principled manner. Since our objective is alignment at the token level rather than hidden-state matching, we removed the feature loss so that training focuses solely on aligning output tokens. We will incorporate these explanations when updating the manuscript.
>
> 2. TALF precomputation cost and generalization
>
> The preprocessing stage of TALF, where the target model first performs tree-based inference on the training dataset to generate target labels, requires a substantial amount of computation. Since the target model is roughly 10--20 times larger than the draft model, this preprocessing step needs about 5--10 times as much computation as one epoch of draft model training. However, because training is carried out over more than ten epochs, we treat the preprocessing time as amortized and negligible in practice.
>
> We used the ShareGPT dataset for training, which is sufficiently general to make the draft model work well under a wide variety of input distributions. In particular, because the draft model is trained by distilling knowledge from the target model, it appears to perform robustly even on inputs drawn from distributions that were not explicitly seen during training.
>
> 3. Interpretation of the SALF threshold
>
>  The SALF threshold has a simple cost benefit interpretation that we will make more explicit. At each expansion step, SALF compares the expected probability gain from growing a branch against the additional computation cost the draft model must pay to produce and verify those extra draft tokens. The threshold corresponds to the point where the expected reduction in target model computation from potential acceptances becomes smaller than the draft side overhead required to explore that branch. When the expected gain falls below the threshold, SALF stops expanding that branch because further drafting is no longer cost effective. Seen this way, the threshold directly encodes how aggressively the system trades additional draft work for possible savings in target decoding, which is intuitive to tune in real systems.
>
>
> 4. Adaptive variants of the SALF threshold
>
>  We have not yet implemented an adaptive SALF threshold that changes during generation based on runtime statistics, but we agree that this is a promising extension. One could, for example, adjust the effective threshold based on observed acceptance rates, draft model confidence or current batch size, in order to dynamically balance quality and overhead across different stages of a response or across tasks. Our current results already show that a single fixed threshold in a broad range gives robust improvements across models and benchmarks, which is why we focused on the fixed setting in the paper.

---

### Official Review · Reviewer_1He7 · 2025-10-28

**Soundness:** 3
**Presentation:** 2
**Contribution:** 3
**Rating:** 6
**Confidence:** 5

**Summary:**

This paper introduces TALF, a tree-aware training loss that aligns a draft model’s predictions with the target LLM across all nodes of a dynamically generated tree, and SALF, an inference-time tree-construction algorithm that stops expanding branches once the expected probability gain falls below a threshold; together they eliminate training-inference misalignment and cut drafting overhead, yielding 15–39 % end-to-end speedups over state-of-the-art speculative decoders without harming output quality.

**Strengths:**

1. The proposed TALF incorporates tree structure into training, improving alignment across all branches, especially low-probability ones.
2. SALF reduces drafting overhead by early stopping, improving end-to-end latency without significantly hurting acceptance length.
3. Together, the TALF and SALF work well with different LLMs (e.g., Llama-2, Llama-3, DeepSeek) and tasks (e.g., MT-bench, HumanEval, GSM8K).

**Weaknesses:**

1. As mentioned in Line.245, the target model is employed to fix the tree structure in advance. However, the draft model is used to generate the tree structure in SALF, which will incur the inconsistence between training and inference. The detailed computational cost of SALF is expected to provide and what about updating the tree structure after the training is stable (such as half of the total epochs) if the computational cost is acceptable.
2. SALF uses a manually set threshold (th=0.6 by default), which may need tuning for different models or tasks. In Sec. 4.4, th=0.5 is best for Deepseek-R1-Distill-Llama-8B. If th=0.6 is claimed to be better, proper experiments should be presented.

**Questions:**

Could the proposed method compared to Eagle-3 or combined with Eagle-3?

---

> ### Author Response · Authors · 2025-11-26
> **Response by Authors to 1He7 (1/2)**
>
> We sincerely appreciate your thoughtful review and constructive feedback.
>
> 1. potential inconsistency between training and inference
>
> Thank you for pointing this out. We agree that, during inference, the draft-generated tree may include tokens that do not fully match the target-generated tree. At first glance, this could raise a concern of training-inference inconsistency, as TALF uses trees constructed solely from the target model. However, TALF is explicitly designed to handle this situation. During training, TALF aggregates loss across all nodes in the tree; not only the top-ranked or correct tokens, but also lower-ranked branches that often diverge from the target’s preferred path are aggregated. As shown in Figure 2(b), TALF significantly improves calibration and accuracy on these lower-ranked nodes, which are exactly the types of nodes that the draft is likely to explore when it makes mistakes during SALF.
> In other words, even though the tree is precomputed offline by the target model, it still contains many candidate nodes that the draft model is likely to produce, including less probable ones. TALF encourages the model to assign meaningful probabilities at these nodes, making it robust to draft-model deviations during inference. Finally, while dynamically updating trees during training could be an interesting extension, it would require repeated target-model invocations, significantly increasing the training cost. We believe exploring such extensions would be valuable for future work.
>
> 2. EAGLE-3 compatibility
>
> The contributions of EAGLE-3 are largely orthogonal to our work; our method can be applied to different draft model architectures and larger datasets. Although we were unable to train models on larger datasets as in EAGLE-3 due to the limited training infrastructures we have, we evaluated how an EAGLE-3 model performs trained with TALF. We observed consistent enhancements in mean acceptance length ($\tau$) across all datasets, as shown below.
>
> | Drafter variant              | MT-bench | HumanEval | GSM8k |
> |-----------------------------|:-------------:|:-------------:|:-------------:|
> | EAGLE-3, with feature loss  |        4.63  | 5.03 | 5.44 |
> | EAGLE-3, without feature loss    |        5.09  | 5.55 | 6.00 |
> | EAGLE-3, TALF               |        5.37  | 5.80 |  6.25 |

---

> ### Author Response · Authors · 2025-11-26
> **Response by Authors to 1He7 (2/2)**
>
> 3. the threshold parameter in SALF
>
> We understand the reviewer’s concern that introducing a threshold parameter (th) may appear to reduce usability. However, our experiments indicate that SALF is not highly sensitive to the exact value of th. In practice, values in the range 0.4--0.7 consistently yield strong speedups across all models and tasks, without requiring per-model or per-task tuning.
> The key point is that SALF outperforms beam search and even the optimal tree search with any value in this range, meaning that it is easy to pick a good threshold such as 0.5 or 0.6 without fine tuning.
> We chose 0.6 as the default because it provided the most stable and consistent improvements across the models as shown in Table 4. This demonstrates that SALF is robust rather than sensitive, and performs well even with a single global setting.
>
> Furthermore, we acknowledge the reviewers’ concern that under highly optimized serving frameworks, the speedup from tree-based SpD may diminish or even become negative. We fully agree, and regard this as a general limitation across all tree-based SpD methods rather than a SALF-specific issue. To directly address this concern and demonstrate that SALF is not merely a theoretical concept but a practically deployable technique, we implemented SALF within the vLLM framework by extending it to support tree-based speculative decoding, which we refer to as vLLM-SpD.
> Unlike prior implementations based on the HuggingFace Transformers library, this integration is closer to practical deployment environments. Using this setup, we show that SALF allows for simple and effective threshold tuning, achieving consistent speedups over existing baselines.
> We conducted experiments on a single A100 GPU. The dataset was constructed by mixing MT-bench, HumanEval, GSM8K, Alpaca, and CNN/DM dataset. We fed requests into the serving framework at a rate of 0.8 requests per second and measured the end-to-end latency (E2E) as the time from when each request was submitted until it completed. The hyperparameters of EAGLE and SALF (top-$k$, depth, and $N$) were tuned to their optimal values under this experimental setup.
> | Method | top-$k$ | Depth | $N$ | Threshold | Mean E2E (s) | p90 E2E (s) | p95 E2E (s) | p99 E2E (s) |
> |:------:|:-------:|:-----:|:---:|:---------:|:-------------:|:------------:|:------------:|:------------:|
> | EAGLE  |   16    |   4   | 60  |     --    |     11.5      |     20.2     |     23.8     |     27.1     |
> | SALF   |   16    |  10   | 60  |   0.0     |     14.9      |     26.1     |     30.5     |     36.5     |
> | SALF   |   16    |  10   | 60  |   0.1     |     12.1      |     20.6     |     24.3     |     30.1     |
> | SALF   |   16    |  10   | 60  |   0.2     |     10.4      |     17.7     |     20.3     |     23.9     |
> | SALF   |   16    |  10   | 60  |   0.3     |     10.2      |     17.1     |     19.8     |     22.5     |
> | SALF   |   16    |  10   | 60  |   0.4     |      **9.72 (-15\%)** |     **16.1 (-20 \%)**    |     **18.1 (-24 \%)**    |     **21.8 (-20 \%)**  |
> | SALF   |   16    |  10   | 60  |   0.5     |     10.0      |     16.6     |     18.3     |     21.7     |
> | SALF   |   16    |  10   | 60  |   0.6     |     10.0      |     17.0     |     18.8     |     22.8     |
> | SALF   |   16    |  10   | 60  |   0.7     |     10.7      |     18.4     |     20.5     |     22.1     |
> | SALF   |   16    |  10   | 60  |   0.8     |     11.1      |     18.6     |     20.6     |     23.9     |
> | SALF   |   16    |  10   | 60  |   0.9     |     12.1      |     20.2     |     22.6     |     25.6     |
>
> The results in the table isolate the impact of the tree construction algorithms in EAGLE and SALF and are therefore independent of TALF. When TALF is additionally applied, the performance gap between baseline and SALF & TALF becomes even larger.

---

### Official Review · Reviewer_Qx4H · 2025-11-03

**Soundness:** 2
**Presentation:** 3
**Contribution:** 2
**Rating:** 2
**Confidence:** 3

**Summary:**

The paper introduces an inference-aware drafter training method by supervision on the tree-decoded tokens of a drafter to align with the target model. Specifically, authors propose a new loss function (TALF) which is a cross-entropy sum for all tree nodes of the drfater model and the new tree-construction mechanism (SALF) which imposes conditional stopping-criterion for reducing the drafter overhead.

**Strengths:**

* The paper introduces inference-aware training of the drafter, which is novel and makes sense in SD literature given that latest method often depends on tree-decoding of the drafter.
* Experiments are conducted with multiple models and datasets and results are solid.
* Presentation is clear and ablation are properly studied.

**Weaknesses:**

* **More advanced baselines** : Authors compare the TALF & SALF with EAGLE-2 and HASS. However, more recent methods like EAGLE-3 [1] improves the performance of EALGE-2 by a large margin, so the proposed method should be compared or combined with [1] ([1] also removes feature alignment loss which alignes with the argument in ln 254).

* **Experiment details** : Some of the experiment setting is unclear or not fair. In ln 352, why taking different approaches for llama series and. Moreover, performance improvement along trained token numbers is lacking.

* **Hyper-parameter sensitivity** : While the authors conducted some ablations on hyper-parameters, naive grid search for SALF threshold (Table 4) and choosing N, B in inference stage weakens the practicality of the algorithm in real serving scenario.

**Questions:**

* What's the size of the drafter models for the experiments? Can author provides the effects of size of the drafter?

* Can author show the scaling effect of the new training algorithm as in EAGLE-3 [1]?

* Can author provide experiment results on other GPU type if possible?

* How does the hyper-parameters N, B, $\tau$ are selected?

[1] (Li et al.) EAGLE-3: Scaling up Inference Acceleration of Large Language Models via Training-Time Test

---

> ### Author Response · Authors · 2025-11-26
> **Response by Authors to Qx4H (1/3)**
>
> Thank you for the detailed and constructive review.
>
> 1. Effect of draft model size and EAGLE-3 compatibility
>
> The main differences introduced in EAGLE-3, relative to EAGLE-2, are (1) the removal of the feature regression loss ($\mathcal{L}_\mathrm{reg}$) and (2) the adoption of a higher-capacity draft model with an additional projection layer. We added comparisons between the EAGLE-3 loss function (1) and TALF.
>
> Also, although the draft model architecture (2) is orthogonal to our contributions, we trained the EAGLE-3 draft model with TALF and other loss functions for comparison. As shown below, TALF achieves consistent improvements in mean acceptance length ($\tau$) when combined with the optimal tree construction algorithm. These results illustrate that TALF is readily extensible to different draft model architectures.
>
>
> | Drafter variant              | MT-bench | HumanEval | GSM8k |
> |-----------------------------|:-------------:|:-------------:|:-------------:|
> | EAGLE-3, with feature loss  |        4.63  | 5.03 | 5.44 |
> | EAGLE-3, without feature loss    |        5.09  | 5.55 | 6.00 |
> | EAGLE-3, TALF               |        5.37  | 5.80 |  6.25 |
>
> 2. Experimental details
>
> For LLaMA-2 and LLaMA-3, we compare HASS and TALF using the same number of training epochs. We agree that one epoch of TALF training takes longer than one epoch of HASS, which may raise concerns about fairness. To address this issue, we conducted an additional experiment on DeepSeek-R1 with a different setup; we trained the draft model using each loss function for the same wall-clock time and then compared the results. Even under the same training time (merely 24 hours in a two-GPU system), TALF still outperforms HASS.

---

> ### Author Response · Authors · 2025-11-26
> **Response by Authors to Qx4H (2/3)**
>
> 3. Hyperparameter sensitivity
>
> For evaluation, we adopted the hyperparameter settings (top-$k$, depth, and $N$) from EAGLE. We agree that, in real systems, the sensitivity of these hyperparameters can become an important practical concern. To evaluate this sensitivity in realistic serving scenarios, we implemented a tree-based speculative decoding framework with a batching support on top of vLLM, which we refer to as vLLM-SpD. vLLM-SpD uses a Triton kernel to implement FlashAttention with a tree attention mask. vLLM-SpD can be made faster by replacing the Triton kernel with a C++/CUDA-based FlashAttention implementation.
>
> In our measurements, vLLM-SpD runs EAGLE-based speculative decoding $3.12\times$ faster than vanilla vLLM and $2.02\times$ faster than vLLM with standard sequential speculative decoding. Within vLLM-SpD, we find that it is easy to identify SALF hyperparameters that run faster than algorithms using the EAGLE tree. This well-optimized framework enables us to conduct experiments that more closely reflect real-world serving conditions.
>
> Regarding the concern that naive grid search for the SALF threshold (th) and the choice of hyperparameters weakens practicality, we view this from the opposite perspective. In a real serving system, what matters is whether reliable hyperparameters can be identified with a simple coarse tuning procedure. Our experiments show that SALF has a wide range of thresholds that outperform the EAGLE-style tree across different workloads (Table~4 in the paper), so even naive grid search quickly finds a good setting that performs robustly across the workloads. We therefore believe that the fact that SALF works well under a simple naive grid search actually supports its practicality in real serving scenarios rather than weakening it.
>
> We conducted vLLM experiments on a single A100 GPU. The dataset was constructed by mixing MT-bench, HumanEval, GSM8K, Alpaca, and a CNN/DM dataset. Requests were fed into the serving framework at a rate of 0.8 requests per second, and we measured the end-to-end latency (E2E) as the time from when each request was submitted until it completed. The hyperparameters of EAGLE and SALF (top-$k$, depth, and $N$) were tuned to their optimal values under this experimental setup.
>
> | Method | top-$k$ | Depth | $N$ | Threshold | Mean E2E (s) | p90 E2E (s) | p95 E2E (s) | p99 E2E (s) |
> |:------:|:-------:|:-----:|:---:|:---------:|:-------------:|:------------:|:------------:|:------------:|
> | EAGLE  |   16    |   4   | 60  |     --    |     11.5      |     20.2     |     23.8     |     27.1     |
> | SALF   |   16    |  10   | 60  |   0.0     |     14.9      |     26.1     |     30.5     |     36.5     |
> | SALF   |   16    |  10   | 60  |   0.1     |     12.1      |     20.6     |     24.3     |     30.1     |
> | SALF   |   16    |  10   | 60  |   0.2     |     10.4      |     17.7     |     20.3     |     23.9     |
> | SALF   |   16    |  10   | 60  |   0.3     |     10.2      |     17.1     |     19.8     |     22.5     |
> | SALF   |   16    |  10   | 60  |   0.4     |     **9.72 (-15\%)** |     **16.1 (-20 \%)**    |     **18.1 (-24 \%)**    |    **21.8 (-20 \%)**  |
> | SALF   |   16    |  10   | 60  |   0.5     |     10.0      |     16.6     |     18.3     |     21.7     |
> | SALF   |   16    |  10   | 60  |   0.6     |     10.0      |     17.0     |     18.8     |     22.8     |
> | SALF   |   16    |  10   | 60  |   0.7     |     10.7      |     18.4     |     20.5     |     22.1     |
> | SALF   |   16    |  10   | 60  |   0.8     |     11.1      |     18.6     |     20.6     |     23.9     |
> | SALF   |   16    |  10   | 60  |   0.9     |     12.1      |     20.2     |     22.6     |     25.6     |
>
> The results in the table isolate the impact of the tree construction algorithms in EAGLE and SALF and are therefore independent of TALF. When TALF is additionally applied, the performance gap between baseline and SALF & TALF becomes even larger.

---

> ### Author Response · Authors · 2025-11-26
> **Response by Authors to Qx4H (3/3)**
>
> 4. Experiments on other GPUs
>
> We conducted additional experiments on a single RTX 4090 GPU. Compared to the A100 GPU, the RTX 4090 has a lower compute capability and more limited memory space, so we reduced the request rate to 0.5 requests per second in our experiments.
>
> | Method | top-$k$ | Depth | $N$ | Threshold | Mean E2E (s) | p90 E2E (s) | p95 E2E (s) | p99 E2E (s) |
> |:------:|:-------:|:-----:|:---:|:---------:|:-------------:|:------------:|:------------:|:------------:|
> | EAGLE  |   16    |   4   | 60  |     --    |     10.2      |     18.9     |     20.8     |     24.5     |
> | SALF   |   16    |  10   | 60  |   0.0     |     12.6      |     24.5     |     27.2     |     30.8     |
> | SALF   |   16    |  10   | 60  |   0.1     |     10.8      |     20.6     |     23.2     |     26.9     |
> | SALF   |   16    |  10   | 60  |   0.2     |     9.69      |     17.8     |     19.9     |     24.0     |
> | SALF   |   16    |  10   | 60  |   0.3     |     9.21      |     17.4     |     19.0     |     22.7     |
> | SALF   |   16    |  10   | 60  |   0.4     |      **9.16 (-10\%)** |     **16.9 (-11 \%)**    |     **18.6 (-11 \%)**    |     **22.4 (-9 \%)** |
> | SALF   |   16    |  10   | 60  |   0.5     |     9.45      |     17.6     |     19.2     |     23.5     |
> | SALF   |   16    |  10   | 60  |   0.6     |     9.77      |     18.1     |     20.1     |     24.2     |
> | SALF   |   16    |  10   | 60  |   0.7     |     10.8      |     20.4     |     22.4     |     27.1     |
> | SALF   |   16    |  10   | 60  |   0.8     |     12.6      |     23.5     |     26.3     |     30.5     |
> | SALF   |   16    |  10   | 60  |   0.9     |     17.7      |     32.1     |     35.2     |     40.5     |
>
> 5. Scaling effect
>
> The scaling effect in EAGLE-3 is achieved through architectural modifications and the removal of feature loss. In contrast to feature-loss-based objectives, TALF provides direct token-level supervision that explicitly guides the draft model to handle low-probability paths--an aspect that EAGLE-3 still does not address.
>
>
> We therefore expect our methods to similarly benefit from this scaling effect and potentially achieve even greater performance under large-scale settings with expanded datasets. However, due to limited computational resources, conducting such large-scale experiments is currently infeasible for us.

---

### Meta-Review · Area_Chair_bQEP · 2026-01-07

**Summary:**

This paper proposes two complementary techniques for tree-based speculative decoding: (1) TALF, a tree-aware loss that aligns the draft model with the target model across all nodes of a speculative tree, and (2) SALF, an inference-time tree construction strategy that adaptively stops branch expansion based on expected probability gain to reduce drafting overhead. Together, these methods aim to resolve a key training–inference mismatch in prior tree-based speculative decoding approaches and demonstrate meaningful end-to-end latency improvements without degrading output quality.

Overall, reviewers agree that the paper addresses a well-motivated and timely problem in speculative decoding. The novelty of inference-aware drafter training (TALF) and the practicality of SALF are widely acknowledged as strengths. However, reviewer opinions diverge on the strength of the contribution relative to the current SOTA and on the sufficiency of experimental validation, leading to mixed scores ranging from rejection to marginal acceptance.

Main Concerns:

1. Comparison with stronger baselines: Multiple reviewers noted the lack of initial comparison to EAGLE-3. While this was a significant concern in the original submission, the authors’ rebuttal provides additional results showing that TALF remains beneficial when combined with EAGLE-3-style draft models.
2. Hyperparameter sensitivity: Reviewers expressed concern about the manual tuning of SALF thresholds and inference parameters. The rebuttal argues convincingly that SALF is robust over a wide threshold range and that coarse tuning suffices in practice.
3. Training–inference consistency and cost: Questions were raised about using target-model-generated trees for training while relying on draft-generated trees at inference, as well as the computational cost of TALF preprocessing. The authors clarify that TALF explicitly trains on low-probability branches, improving robustness to draft deviations, and that preprocessing cost is amortized over training.
4. Presentation and clarity: Some reviewers felt that certain design choices (e.g., removal of feature loss, interpretation of the SALF threshold) were under-explained in the paper and would benefit from clearer justification.

The authors’ rebuttal addresses some of the concerns, such as parameter sensitivity. Some limitations remain, such as the lack of large-scale scaling experiments comparable to EAGLE-3. Taking all reviews and the rebuttal into account, this submission sits near the margin and would benefit from clearer exposition and tighter positioning against the strongest existing baselines.

**Reviewer Concerns:**

see above

**Reviewer Scores:**

Reviewer opinions diverge on the strength of the contribution relative to the current SOTA and on the sufficiency of experimental validation, leading to mixed scores ranging from rejection to marginal acceptance. Although the rebuttal addresses some of the concerns, I still feel the paper would benefit from another round of major revision before it can be published.

---

### Decision · Program_Chairs · 2026-01-26

Reject